# Atomic force microscopy reveals structural variability amongst nuclear pore complexes

George J Stanley[1], Ariberto Fassati[2], Bart W Hoogenboom[1,3]

**The nuclear pore complex (NPC) is a proteinaceous assembly that regulates macromolecular transport into and out of the nucleus. Although the structure of its scaffold is being revealed in increasing detail, its transport functionality depends upon an assembly of intrinsically disordered proteins (called FG-Nups) anchored inside the pore's central channel, which have hitherto eluded structural characterization. Here, using high-resolution atomic force microscopy, we provide a structural and nanomechanical analysis of individual NPCs. Our data highlight the structural diversity and complexity at the nuclear envelope, showing the interplay between the lamina network, actin filaments, and the NPCs. It reveals the dynamic behaviour of NPC scaffolds and displays pores of varying sizes. Of functional importance, the NPC central channel shows large structural diversity, supporting the notion that FG-Nup cohesiveness is in a range that facilitates collective rearrangements at little energetic cost. Finally, different nuclear transport receptors are shown to interact in qualitatively different ways with the FG-Nups, with particularly strong binding of importin-$\beta$.**

## Introduction

The nuclear pore complex (NPC) is a selective, nanoscale filter for macromolecules entering or exiting the nucleus. It is a large, proteinaceous assembly spanning the nuclear envelope (NE), comprising ~30 different nuclear pore proteins (Nups), each present in many copies to give a total of ~500–1,000 Nups (Reichelt et al, 1990; Cronshaw et al, 2002; Bui et al, 2013; Kim et al, 2018). The scaffold of the NPC has an eightfold rotational symmetry and is formed from three intercalating ring structures: the cytoplasmic ring complex, the inner ring complex, and the nucleoplasmic ring complex (Eibauer et al, 2015; Von Appen et al, 2015; Kosinski et al, 2016; Kim et al, 2018). With a scaffold diameter of ~90 nm, a central channel diameter of ~40 nm, and a mass of ~60–125 MD (dependent upon species), the NPC constitutes one of the largest protein complexes

in the eukaryotic cell. Its central channel is occluded by natively disordered proteins, called FG-nucleoporins (or FG-Nups), which, in conjunction with soluble nuclear transport receptors (NTRs), form a selective barrier to transport. This barrier is selective for molecules as small as ~4–5 nm in diameter (Mohr et al, 2009; Schmidt & Görlich, 2016; Timney et al, 2016), and yet still permeable to others as large as ~39 nm (including human hepatitis B virus capsids) (Panté & Kann, 2002). Furthermore, it deals with traffic travelling in both directions simultaneously, with chemically very divergent cargo molecules (from RNA to proteins to viruses), and it does all this very quickly: it has been estimated that a mass of nearly 100 MD translocates a single NPC every second (Ribbeck & Görlich, 2001). And yet, although our understanding of the NPC's structural scaffold is being revealed in ever increasing detail, the functionally most relevant part of the NPC, the central transport barrier, remains poorly characterized at best (Eibauer et al, 2015; Von Appen et al, 2015; Kosinski et al, 2016; Kim et al, 2018). This is presumably due to its disordered nature and the presence of macromolecules trapped inside the central channel, both of which make it less amenable to the ensemble averaging procedures used in many structural techniques (recent electron microscopy studies have either removed the transport barrier for the averaging procedure or shown it as a two-lobed blur [Eibauer et al, 2015; Kim et al, 2018]). However, because the nucleocytoplasmic filtering mechanism resides in this FG-Nup assembly, our understanding of the NPC would benefit from a more thorough characterization of the FG-Nup morphology inside the central channel.

The morphology of these FG-Nups has traditionally been hypothesized as something resembling a diffuse polymer brush (Timney et al, 2016), or a dense hydrogel (Hülsmann et al, 2012) (for a review, see Stanley et al [2017]). However, more recent studies of FG-Nup behaviour in planar films, coupled with computational modelling work, have suggested that FG-Nups demonstrate a balance of both these two extreme behaviours (Vovk et al, 2016; Zahn et al, 2016), with a possible propensity towards gel formation (Ghavami et al, 2018). Furthermore, from a nanomechanical characterization of intact NPCs, supported by computational modelling, it has been predicted that the cohesive properties of the FG-Nups lie in a range that facilitates transitions between different (meta-)

[1]London Centre for Nanotechnology, University College London, London, UK [2]Division of Infection and Immunity, University College London, London, UK [3]Department of Physics and Astronomy, University College London, London, UK

Correspondence: b.hoogenboom@ucl.ac.uk

stable collective morphologies (Osmanović et al, 2012; Bestembayeva et al, 2015). This would allow the collective rearrangement of FG-Nups at little energetic cost, thus enabling both the passage of large cargoes and the fast resealing of the barrier. Consistent with this prediction, slowly transitioning "clumped" morphologies have recently been observed for reconstituted FG-Nups inside synthetic NPC mimics (Fisher et al, 2018). However, there remains a need for an experimental method that can directly visualize the collective behaviour of FG-Nups inside a single, intact NPC. With a high signal-to-noise ratio, no chemical tagging or fixation, and the ability to image in solution at ambient temperatures, atomic force microscopy (AFM) is one such method.

Although previous AFM imaging studies have shown the presence of a "central plug" (Stoffler et al, 2003), and possibly the fluctuations of FG-Nups inside NPCs (Sakiyama et al, 2016), the spatial resolution has hitherto been too low to visualize the collective behaviour of FG-Nups inside the central channel. In this study, taking advantage of enhanced control of the probe–sample interaction, reproducibly sharper AFM probes, and increased speed of data acquisition, we apply high-resolution AFM and a fast force-spectroscopy technique to isolated *Xenopus laevis* oocyte NEs, revealing detail at the membrane that has never before been captured. Of functional importance, we show variability amongst the NPCs' central channel, consistent with the prediction that FG-Nup cohesiveness lies within a certain range such that the FG-Nups can alternate between different collective rearrangements at little energetic cost.

## Results

### High-resolution AFM imaging of the NE

Over the past two decades, AFM has been extensively applied to isolated NEs (Oberleithner et al, 1994; Jäggi et al, 2003; Stoffler et al, 2003; Kramer et al, 2008; Bestembayeva et al, 2015; Liashkovich et al, 2015; Sakiyama et al, 2016; Mohamed et al, 2017). During that period, AFM technology has developed new imaging modes, better control of the tip–sample interaction, and improved probe sharpness and consistency, facilitating AFM experiments on soft, biological samples in solution. Here, we exploit these developments for imaging NEs at previously unattainable spatial resolution and reproducibility. This is exemplified on the cytoplasmic side of an NE (Fig 1A) that was mechanically isolated from a *X. laevis* oocyte (see the NE preparation for AFM imaging section). As probed here with AFM tips of 2-nm nominal radius and tip half opening angle of ≤20° (MSNL-F; Bruker; see the Atomic force microscopy section) and consistent with their appearance in previous AFM experiments (Oberleithner et al, 1994; Jäggi et al, 2003; Stoffler et al, 2003; Kramer et al, 2008; Bestembayeva et al, 2015; Liashkovich et al, 2015; Sakiyama et al, 2016; Mohamed et al, 2017), NPC scaffolds stand out against the NE as ring shapes of 87 ± 4 nm in diameter (defined as the highest point to highest point of the scaffold structure; $n = 583$; see Fig S1). Spindly protrusions are sometimes seen emanating from the scaffold structure, presumably representing the cytoplasmic filaments. In some cases, these appear to bind one NPC to another (see white

arrows in Fig 1A); this connection between NPCs is also consistent with the finding that some messenger ribonucleoproteins "scan" the cytoplasmic periphery before exiting into the cytoplasm (Smith et al, 2015). Strikingly, each NPC has a unique appearance—particularly in the pore lumen. This observation is highlighted in Fig 1B–G; furthermore, these results are robust (see Fig S2, which shows that trace and retrace images of the same pores are slightly shifted with respect to each other because of scanner hysteresis, but are otherwise not significantly different). This structural variability is dynamically stable (and reproducible), as demonstrated by repeat measurements of the same pores imaged after 17 min (Fig S3). Some pores (e.g., Fig 1B) contain a central protrusion that may be attributed to cargo stuck in transit (Stoffler et al, 2003; Sakiyama et al, 2016). Other NPCs (e.g., Fig 1D and F) display structures spanning the lumen, reminiscent of FG-Nups condensing in the centre of the channel, and other NPCs (e.g., Fig 1C) show dense structures near the pores' inner walls, consistent with local condensation of FG-Nups as may result from their cohesive interactions in a confined environment (Osmanović et al, 2012)— behaviour that has recently been observed in NPC mimics containing reconstituted FG-Nups (Fisher et al, 2018).

The enhanced spatial resolution and fidelity of our AFM images is further exemplified at the nucleoplasmic side of the NE (Fig 1H and I). It is distinguishable from the cytoplasmic side by the presence of diffuse protrusions on the NPCs, which at a higher magnification (Fig 1J–L)—and without need for fixation—can be identified as the nuclear baskets. In addition, these AFM images reveal a network of tightly bunched filaments, with little or no spacing between them, running in tandem around the NPCs (see white arrows in Fig 1H for examples). In size and appearance, these resemble electron microscopy observations of the lamin protofilaments comprising the lamina network (Aebi et al, 1986; Turgay et al, 2017), not previously observed by AFM. In Fig 1I, this meshwork appears stretched in comparison with Fig 1H. This may be caused by different mechanical strain applied to the NE during sample preparation. Larger filaments (stretching over 100 s of nm; diameter of ~9 nm: see Fig S4) interweave around and above the NPCs. They sometimes branch and appear to anchor to the NE (see Fig 1H, inset). Given their widths and lengths, as well as their apparent anchoring to the NE, these are likely actin filaments (see Fig S4). The NPC shown in Fig 1L is unusually large, with an estimated scaffold diameter (and hence circumference) consistent with a ninefold instead of the usual eightfold rotational symmetry of the NPC (Hinshaw & Milligan, 2003; Löschberger et al, 2014). Further analysis of NPC sizes shows a Gaussian distribution centred around a radius of 44 nm (consistent with an eightfold rotational symmetry; see Fig S1). In a sample of 583 imaged NPCs, 16 are found to have a radius less than 39 nm, and 10 to have a radius greater than 49 nm—these are likely NPCs with sevenfold and ninefold rotational symmetry, respectively.

Taken together, these data reveal striking heterogeneity and variability at both sides of the NE, revealing its overall organization at unprecedented spatial resolution. Of particular physiological interest is the variability observed inside the NPC lumens: it is consistent with computational predictions of FG-Nups facilitating nucleocytoplasmic transport by alternating between different condensed or clumped collective arrangements (Osmanović et al, 2012, 2013). These are expected to give rise to areas of different local density and surface structure within the NPC lumen, as observed here. Such conformational variability mostly eludes microscopy

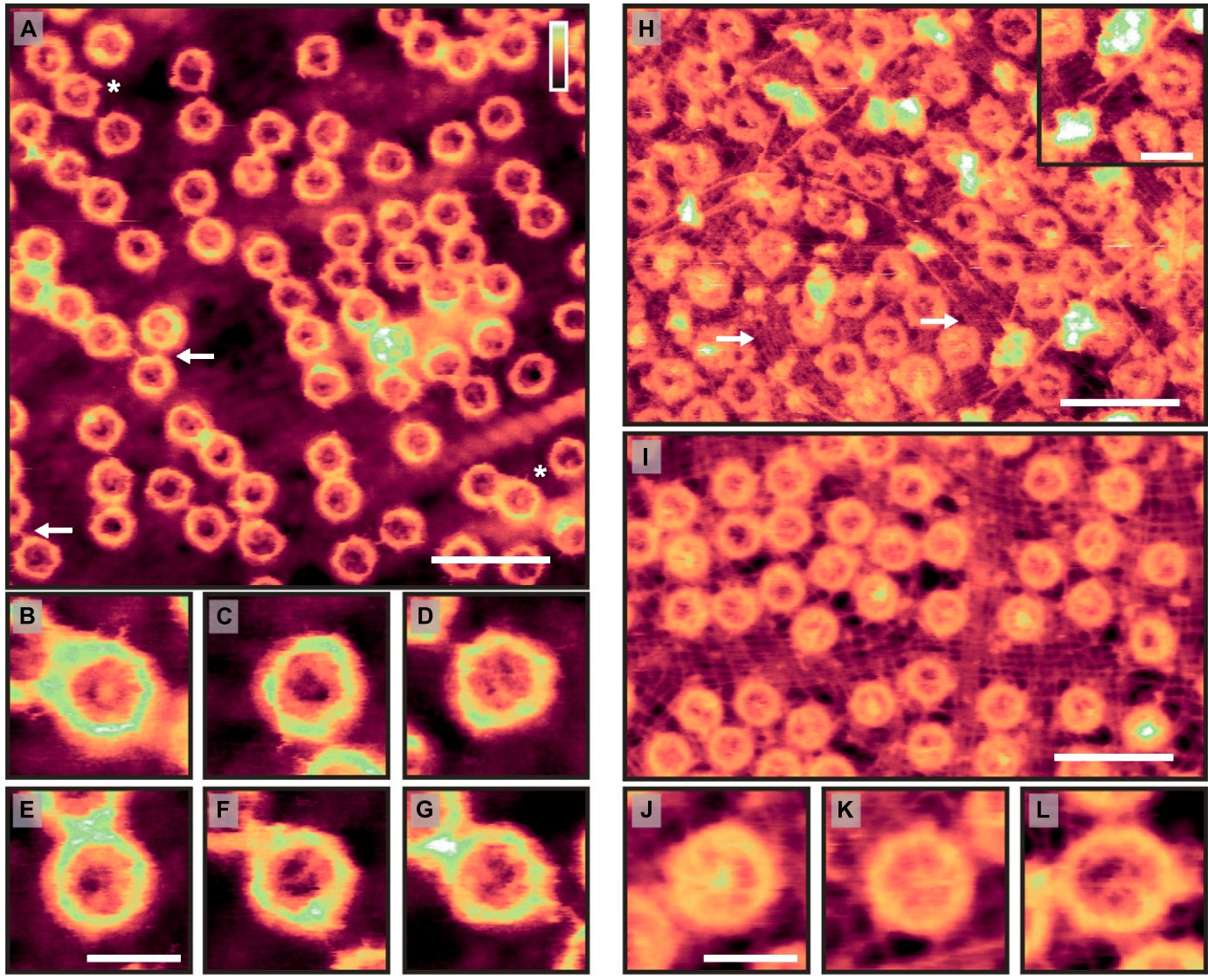

**Figure 1. High-resolution AFM imaging of intact *X. laevis* oocyte NEs in solution.**
**(A)** AFM topography of the cytoplasmic side of the NE. White asterisks denote two (out of several) possible appearances of cargo molecules stuck in transit (see the High-resolution AFM imaging of the NE section). The white arrows show instances of NPCs connecting to one another—likely by their cytoplasmic filaments. **(B–G)** Magnified views of NPCs highlighting the observed variability in the pore lumens. **(H)** Nucleoplasmic side of the NE. The lamina meshwork is observed as tightly bunched filaments running in tandem around the NPCs, with little or no spacing between them (white arrows show patches of exposed lamin protofilaments). In addition, there are longer filaments (presumably actin, see Fig S4) that interweave around the NPCs, sometimes branching. Inset: apparent branching and termination—and possibly anchoring—of such filaments on the NE. **(I)** As (H), but with the lamina meshwork appearing more stretched. **(J–L)** Higher magnification images of NPCs, revealing spoked structures consistent with the nuclear basket. The NPC in (L) is unusually large with a scaffold diameter of 100 ± 4 nm: larger than the usual measured diameter of 85 ± 4 nm (*n* = 282 for nucleoplasmic NPCs; see also Fig S1). Scale bars: 300 nm (A, H, I); 100 nm (B–G; H, inset; and J–L). Colour scales (height, see top right in A): 100 nm (A, H, I), 70 nm (H, inset), 60 nm (B–G), and 65 nm (J–L).

methods that heavily rely on ensemble averaging for obtaining nanometre-range resolution (Eibauer et al, 2015; Kim et al, 2018).

## High-throughput nanomechanical characterization of the NE

Better interpretation of the AFM images would require molecular identification and information that extends below the top surface of the NPC. Whereas chemically specific AFM methods do not yet provide the required resolution, further information is readily accessible via the (nano)mechanical properties of the sample. Such properties can be inferred from force-spectroscopy measurements, which record the forces required to locally indent the sample with the AFM tip. This approach was previously used to probe NPC lumens—and thus the transport barrier—to depths exceeding 20 nm (Bestembayeva et al, 2015). With the increase in data acquisition speed, it is now possible to acquire force curves at the order of kHz (number of force versus distance curves per second), enhancing the throughput of such measurements by two orders of magnitude and allowing the force curves to be acquired while recording images at a similar spatial resolution as that demonstrated in Fig 1.

Higher acquisition speed requires a larger measurement bandwidth, which introduces more noise and therefore makes it harder

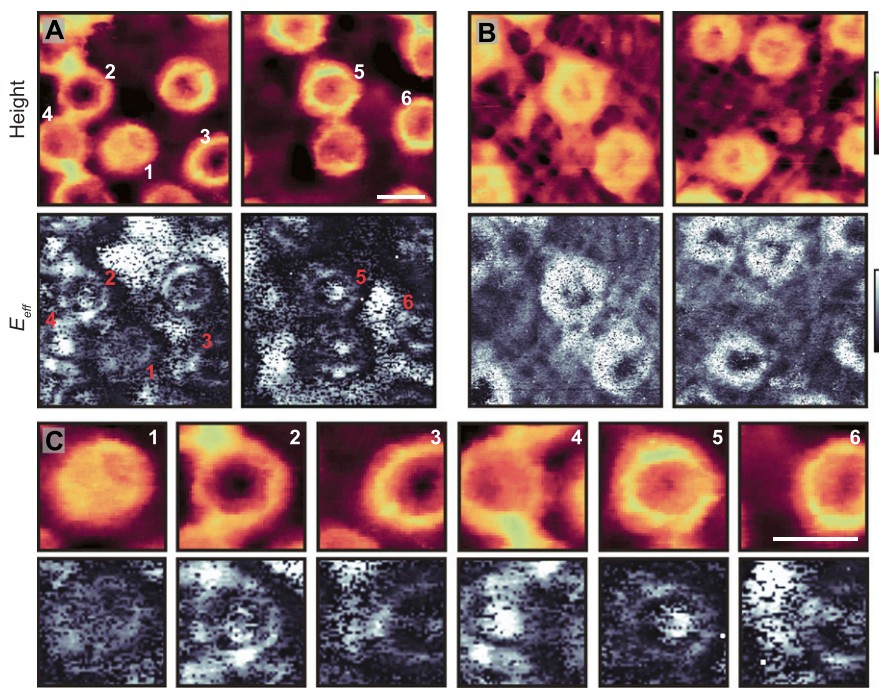

**Figure 2. Nanomechanical characterization of the NE.**
**(A)** Two images from the same sample showing the cytoplasmic side of the NE (top row), with the effective elastic moduli ($E_{eff}$)—determined from Hertz fits to individual force curves captured during imaging (bottom row; force curve frequency: 2 kHz). **(B)** As (A), for the nucleoplasmic side of the membrane, highlighting the NPC scaffolds and the lamina network as local enhancements in height and $E_{eff}$, and NPC baskets as local increases in height and as reductions in $E_{eff}$. **(C)** Cropped pores from (A), highlighting the variability in transport channel $E_{eff}$ values for NPCs with similar topographies. Force at maximum indentation ($F_{max}$, ±10% confidence interval): 397 pN (A) and 300 pN (B). Scale bar for all images (see A and C, top right panels): 100 nm. Colour scales: 70 nm and 6 MPa (A and C); 75 nm and 3 MPa (B).

to detect the point of contact between tip and sample—particularly when probing soft materials such as the NE. In addition, the higher indentation rate, and therefore greater tip velocity, is likely to lead to a stiffening of the sample in response to indentation. These effects can be demonstrated by comparing the high-speed force-spectroscopy results with the conventional, low-throughput experiments—both on the cytoplasmic and nucleoplasmic side of the NE—on ensemble-averaged data sets (see the Validating fast force-spectroscopy methods: PeakForce QNM section and Fig S5). As expected, the NPCs appear stiffer with the higher throughput method. This is clear from the smaller indentation for (approximately) the same applied force, and from the higher effective elastic moduli ($E_{eff}$) as determined by fitting the force curves with a Hertz indentation model. It is noted that such an indentation model should be considered purely phenomenological in the context of the NPC, that is, the resulting elastic moduli are not viable as absolute measures, but still useful for comparative measurements. Notwithstanding the quantitative differences, the qualitative features are conserved between both methods: enhanced stiffness at the NPC scaffold structure and in the centre of the pore lumen for the cytoplasmic side (Bestembayeva et al, 2015), and, on the nucleoplasmic side, a local reduction in stiffness in the centre of the pore due to the rather soft/flexible nuclear basket.

With the high-speed data acquisition, these same features can now be observed without ensemble averaging, as here demonstrated with AFM images alongside their concomitant effective elastic moduli (see Fig 2). When imaged from the cytoplasmic side (Fig 2A), the scaffold ring structures are seen to protrude from the membrane, and their $E_{eff}$ gives a sometimes strong (white) response. Furthermore, some NPCs display a firm $E_{eff}$ from their pore lumen (e.g., the pores marked 2, 4, and 5)—suggesting the tip is interacting with a stiff, dense material. The NPCs imaged from the nucleoplasmic side (Fig 2B), however, render soft (dark) $E_{eff}$ values from their pore lumens at the position of the nuclear basket. In addition, as expected, the lamina meshwork stands out not only by its AFM topography but also by its enhanced elasticity with respect to the NE. Interestingly, however, its $E_{eff}$ tends to be less than that of the nucleoplasmic NPC scaffold structures.

Importantly, there is variability between the different NPCs, both in the AFM topography and in the $E_{eff}$ heatmaps. NPC 1 shows a triangular structure protruding over the lumen which gives a soft $E_{eff}$ (Fig 2C). Both NPCs 2 and 3 display gaps in the central channel at the cytoplasmic periphery, and yet they markedly differ in their elastic response with one showing a central stiffness enhancement (2), whereas the other renders a weaker $E_{eff}$ (3). Similarly, pores 4–6 are all occluded at the cytoplasmic periphery, yet only NPCs 4 and 5 give a strong central $E_{eff}$—pore 6 gives a markedly reduced response from the central channel. This suggests that there exists relatively stiff, dense material deeper down inside the pore, which can be more pronounced in some NPCs (e.g., pore 5), but largely absent in others within the same image (pore 6). These signatures typically extend over multiple pixels in the image, hence multiple force curves, and can therefore not be attributed to measurement noise (they are also reproduced with subsequent scans—see Fig S6).

These data demonstrate that the spatial heterogeneities inside NPC lumens extend over many nanometres laterally and vertically into the transport barrier, that is, below the top surface. They should therefore be attributed to collective molecular configurations inside the pores, most likely because of local condensation and rearrangements of FG-Nups and NTRs. (Previous AFM and confocal fluorescence microscopy experiments have excluded trapped cargoes as a significant factor in these measurements [Bestembayeva et al, 2015].)

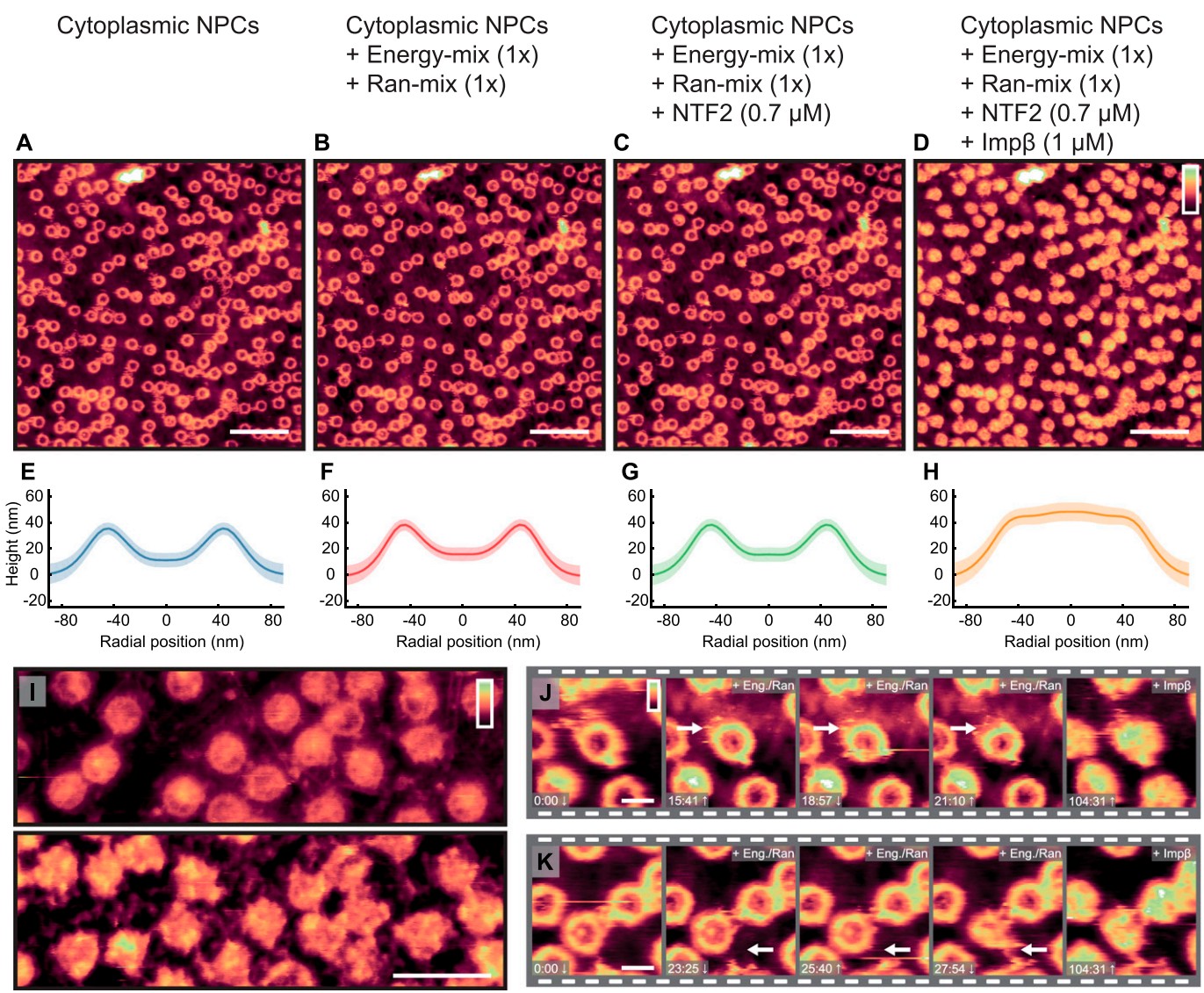

**Figure 3. Effect of NTRs on NPCs.**
**(A–D)** Images from a video sequence (see Video 1) of the cytoplasmic side of the NE, in which, step-by-step, many of the proteins (and chemical energy) required for the classical import cycle of NLS proteins are added to the system. **(A)** Cytoplasmic side of NE. **(B)** After addition of the Ran mix and energy mix. **(C)** hsNTF2 (0.7 μM) is added to the sample. **(D)** hsImpβ (1 μM) is added and all NPCs fill with protein. **(E–H)** The rotationally averaged height profiles of the cross-correlation averaged NPCs from the images displayed in (A–D), respectively, showing a filling of the pore lumen and some increase in the pore rim height upon incubation with hsImpβ. **(I)** Nucleoplasmic side of the membrane before (top) and after (bottom) addition of hsImpβ (1 μM). **(J, K)** Cropped pores from the image sequence (A–D) showing changes as a function of time. Scale bars: 600 nm (A–D), 300 nm (I), and 100 nm (J and K). Colour scales: 150 nm (A–D) and 80 nm (I–K).

## Effects of NTRs on the transport barrier

This raises the question: How do soluble transport receptors affect the various morphologies observed inside the NPC? To address this question, we used high-resolution AFM imaging to visualize any changes to the NPCs upon subsequent addition of Ran- and energy mixes followed by NTRs to thus provide the cytosolic reagents required for specific import. Two different NTRs were tested: *Homo sapiens* nuclear transport factor 2 (hsNTF2), which is a homodimer with a total molecular weight of ~29 kD (Bullock et al, 1996), and has two known FG-Nup–binding sites (Bayliss et al, 2002); and, *Homo sapiens* importin-β (hsImpβ), which is a subunit of the Impα·Impβ

heterodimer, has a molecular weight of ~100 kD, and is estimated to have nine FG-Nup–binding sites (Isgro & Schulten, 2005).

Fig 3A shows the cytoplasmic side of the membrane, in import buffer, before addition of exogenous proteins. Upon addition of the Ran- and energy mixes (Fig 3B), very little change to the structures inside the pore lumens is observed, although some binding of proteins to the NPC scaffold is detected, as well as a slight swelling of the barrier (shown by a very small increase in height in the rotationally averaged plots: compare panels Fig 3E and F). After addition of hsNTF2 (Fig 3C), again very little change is observed; the rotationally averaged plot (Fig 3G) suggests that on average, no significant swelling of the transport barrier occurs. However, after

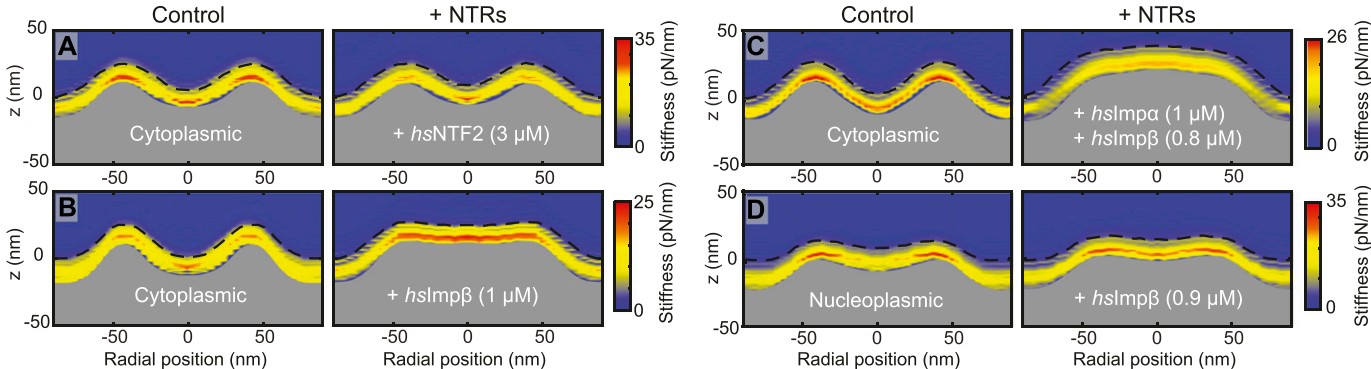

**Figure 4. Differences between importin-*β* and NTF2 binding in the NPC.**
**(A)** Stiffness heatmap of the cytoplasmic NPC surface before (left; *n* = 51) and 5 min after (right; *n* = 24) addition of *hs*NTF2 (3 *μM*). Before addition of *hs*NTF2, stiffness is seen at the cytoplasmic ring structure, and in the central transport barrier (see also Fig S5)—and, after addition of *hs*NTF2, the same pattern is observed. **(B)** Cytoplasmic NPC surface before (*n* = 66) and 5 min after (*n* = 67) addition of *hs*Imp*β* (1 *μM*), showing filling and homogenization of the central channel. **(C)** Cytoplasmic NPC surface before (*n* = 26) and 5 min after (*n* = 41) addition of *hs*Imp*α* (1 *μM*) and *hs*Imp*β* (0.8 *μM*), showing a similar effect. **(D)** Nucleoplasmic NPC surface before (*n* = 34) and 5 min after (*n* = 34) addition of *hs*Imp*β* (0.9 *μM*). Without *hs*Imp*β*, the NPC shows a soft centre because of the presence of the flexible nuclear basket (see also Fig S5). *hs*Imp*β* increases the height profile in the centre of the NPCs and homogenizes the stiffness across the central channel. Each experiment (control and with NTRs) was conducted on one NE. $F_{max}$ (±10% confidence interval): 350 pN (A), 283 pN (B), 300 pN (C), and 397 pN (D).

addition of *hs*Imp*β*, all NPCs fill with protein (Fig 3D and H), and all NPC lumen structures change. After 50 min of imaging, all NPCs remain filled, suggesting that the binding of *hs*Imp*β* to the NPC is stable (the complete video sequence can be seen in Video 1). When *hs*Imp*β* was incubated with the NE (at concentrations as low as 200 n*M*), followed by multiple (6) washing steps with buffer to leave only ~0.1 nM of exogenous protein, it is still observed bound inside the pore lumens (see Fig S7). Furthermore, when *hs*Imp*β* is added to the nucleoplasmic side of the membrane (and the buffer is washed of all exogenous protein), it is still seen to bind to the nuclear periphery of the NPC—likely to the nuclear basket—further indicating that *hs*Imp*β* strongly binds to the NPC (Fig 3I). This stable binding of *hs*Imp*β* is expected because the Ran mix does not contain RanGTP (required for *hs*Imp*β* release and present in the nucleus) and neither can it be generated in our conditions because the required nuclear Ran guanine nucleotide exchange factor (RanGEF – also termed RCC1), is not present in isolated NEs. Attempts to add recombinant RanGTP led to inconclusive results because RanGTP tended to induce a detachment of the NE from the substrate and to contaminate the AFM tip.

In this sequence of images, some NPCs display dynamic behaviour: one appears to undergo a conformational change to the scaffold structure (Fig 3J) and others appear to bind to neighbouring NPCs (Fig 3K). Both events occur in the presence of the energy- and Ran mixes. Although it is not possible to determine from these imaging experiments alone whether these interesting events are of any functional importance, we still note their sightings.

Nanomechanical measurements can again provide information on how far the observed phenomena extend below the NPC surface, following a procedure applied previously (Bestembayeva et al, 2015), but here carried out at enhanced speed. Consistent with the AFM imaging results (Fig 3C and G), the addition of *hs*NTF2 results in little change to the nanomechanical properties of the NPC transport barrier (Fig 4A). On the other hand, after injection of *hs*Imp*β* (Fig 4B), the average transport barrier height increases and the stiffness is homogenized across the pore. A similar effect is observed after addition

of both *hs*Imp*α* and *hs*Imp*β* together (Fig 4C): that is, a considerable swelling of the transport barrier and a smearing of the nanomechanical properties across the pore. Similarly, on the nucleoplasmic side of the NE, when *hs*Imp*β* is added, a swelling in the pore lumen is recorded as well as an increase in its stiffness (Fig 4D).

Both the imaging experiments described previously (see Fig 3) and the stiffness heatmaps displayed here (Fig 4), show that *hs*Imp*β* binds stably to the transport barrier and changes its nanomechanical properties, whereas *hs*NTF2 has no significant effect. It is probable that the smaller *hs*NTF2 is optimized to pass through the NPC very quickly to maintain the RanGDP:RanGTP gradient (its mass of ~29 kD is anyway less than the ~30–40-kD exclusion limit for active transport [Mohr et al, 2009; Schmidt & Görlich, 2016; Timney et al, 2016]), without having to break many FG-Nup·FG-Nup interactions. Larger NTRs, however, will require more (or stronger) FG-Nup–binding interaction sites to penetrate the barrier. As such, they are expected to break FG-Nup·FG-Nup interactions, thereby rearranging the transport barrier—as observed here for *hs*Imp*β* and the *hs*Imp*α*·*hs*Imp*β* heterodimer (and as previously observed with low-throughput nanomechanical measurements for *hs*Imp*β* at the NPC's cytoplasmic periphery [Bestembayeva et al, 2015]).

## Discussion

Our high-resolution AFM imaging and fast force-spectroscopy experiments have revealed the complexity at the NE in astonishing detail. NPCs with sevenfold and ninefold rotational symmetry have been identified (Fig S1), confirming earlier reports of ninefold symmetric NPCs (Hinshaw & Milligan, 2003; Löschberger et al, 2014), and in accordance with fluorescence studies that suggest a certain degree of architectural plasticity (Rajoo et al, 2018). In addition, the images reveal dynamic behaviour of NPC scaffold structures (Fig 3 and Video 1), as well as tightly packed lamin protofilaments and interweaving actin filaments, never before resolved by AFM (Fig 1).

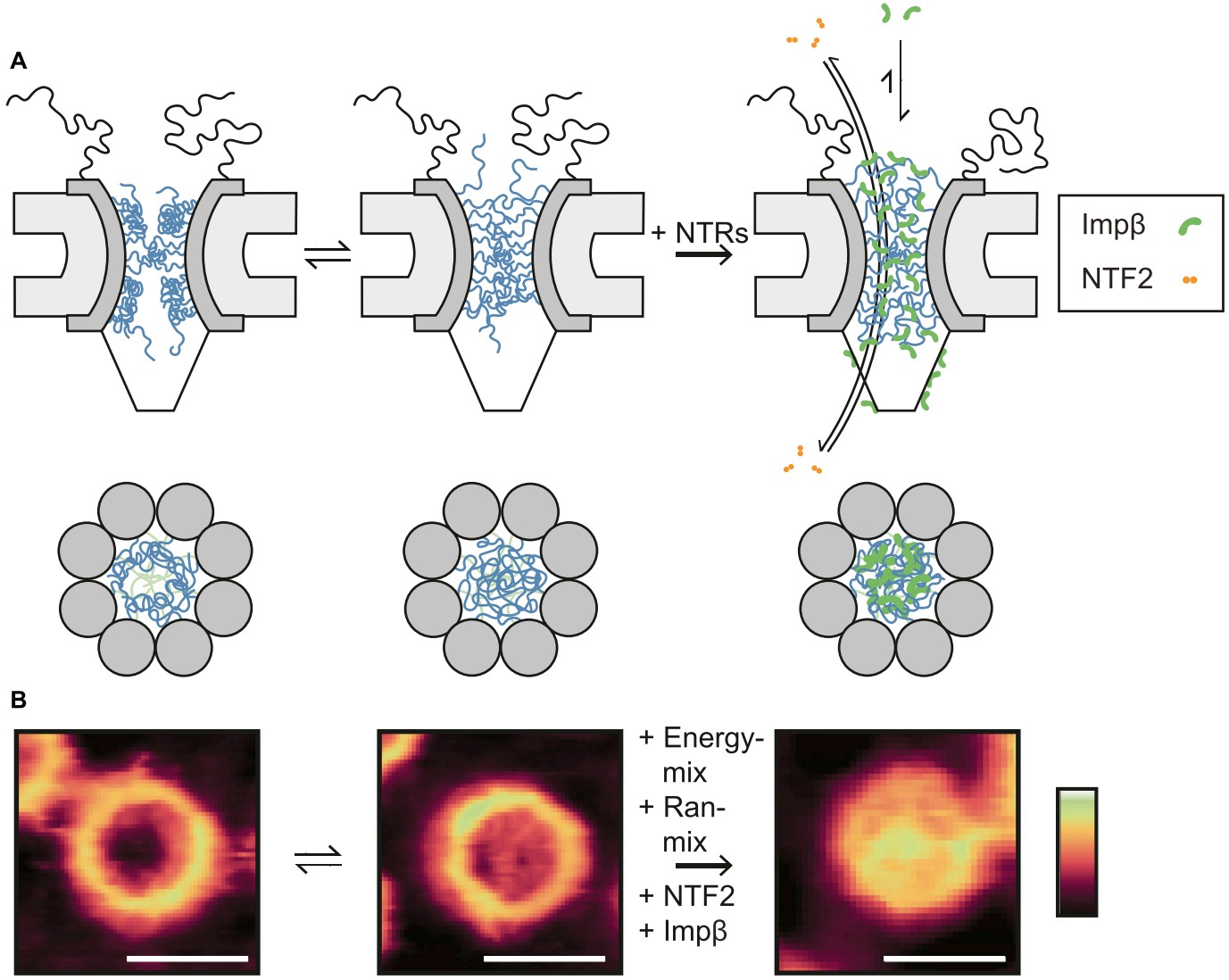

**Figure 5. Proposed role of NTRs in the transport barrier.**
**(A)** In the absence of transport receptors, the FG-Nups can readily alternate between morphologies with enhanced density at the wall (left column) and at the centre (middle column) of the NPC channel. Added transport receptors lead to a homogenization of the FG-Nups across the pore lumen, provided that their binding avidity to the FG-Nups is strong enough (as for *hs*Impβ). Smaller transport receptors, such as NTF2, translocate the transport barrier very quickly without significantly rearranging the FG-Nups. **(B)** Images of NPCs consistent with the different proposed conformations. Scale bars: 100 nm. Colour scales: 70 nm (left and middle image) and 100 nm (right image).

Furthermore, nanomechanical measurements highlight the great strength of the NPC scaffold structure, suggesting that NPCs have a significant role to play in the mechanical stability of the nucleus (Fig 2).

Beyond these observations on the general architecture and organization of the NE and NPC scaffold, high-resolution AFM images demonstrate that no NPC is the same and that differences are particularly noticeable for the functionally most relevant part of the NPC: the disordered FG-Nups of the central channel. There are sometimes ~10–20-nm height variations at various locations inside the NPC pore lumen (Fig S2), going well beyond the presence/absence of a "central plug" as noted in earlier AFM studies (Stoffler et al, 2003; Sakiyama et al, 2016). In fact, the features resolved inside

the NPC lumen are consistent with computational predictions of various metastable FG-Nup conformational states (Osmanović et al, 2012, 2013), and they resemble (in shape and size) the slowly transitioning FG-Nup morphologies recently observed inside mimetic NPCs (Fisher et al, 2018). We therefore interpret these structures as FG-Nups that have condensed into different morphologies, although we note that there will also be a contribution from endogenous NTRs and possibly cargo molecules (Fig 5); a contribution from cytoplasmic filaments cannot be fully excluded either, but is most unlikely given their flexible nature in solution. In this interpretation, some pores show FG-Nups condensed to the inner walls of the channel at the cytoplasmic periphery (Fig 1C), whereas others display a meshwork-type conformation with

FG-Nups occluding the entire channel (also at the cytoplasmic periphery; Fig 1D)—others show a mixture of both (Fig 1F). (We note that, in addition, some more diffuse FG-Nups may still be present above the NPCs—i.e., emanating into the solution—but of too low density/cohesiveness to be detected by AFM.)

The functional relevance of these observations lies in the possibility of FG-Nups to transition between different collective conformations to facilitate transport. The large variety of conformations suggests that there are multiple distinct FG-Nup arrangements. This is consistent with the prediction that the cohesiveness of the FG-Nups is in a range (not too strong, not too weak) that allows transitions between distinct (meta-)stable collective states (as has been postulated in previous theoretical studies [Osmanović et al, 2012]). This would allow large conformational rearrangement at little energetic cost, thus enabling both the passage of large cargoes and the fast resealing of the barrier. In this interpretation, the observed structures are kinetically trapped metastable states of FG-Nup morphologies—and indeed, time-lapse imaging of the same NPCs shows that these structures are static at the timescale of our measurements (Fig S3). Beyond these static features, some dynamics may be present at the ~1-nm length scale (Sakiyama et al, 2016); but in our experiments, any such dynamics do not exceed the measurement noise (Fig S2).

In addition to the FG-Nups, NTRs (especially those of the Impβ family) have been proposed to play an important role in the formation of the transport barrier (Jovanovic-Talisman et al, 2009; Kapinos et al, 2014, 2017; Lowe et al, 2015; Wagner et al, 2015). In fact, they are likely to contribute to the structural variability observed inside the NPC lumens, and they may also account for their static nature, as, even for isolated NPCs, transport receptors and cargos in the central channel may exceed—in terms of mass—the FG-Nups by threefold (Kim et al, 2018). In this study, the addition of exogenous Impβ and the Impα·Impβ heterodimer resulted in filling of the NPCs and homogenization of their nanomechanical properties (Figs 3 and 4). Furthermore, Impβ was hard to remove from the NPC lumens by washing, even after incubation at concentrations as low as 200 nM (Fig S7). This suggests that Impβ binds strongly to the transport barrier, and that therefore some Impβ molecules (and possibly some Impα·Impβ heterodimers) are at all times bound to/in the transport barrier, in accordance with single-molecule fluorescence studies (Lowe et al, 2015). On the other hand, the addition of the smaller NTR, NTF2, did not yield any significant changes to either the observed FG-Nup morphologies or to the averaged nanomechanical properties of the transport barrier (Figs 3 and 4). It should be noted, however, that our measurements were taken in the absence of a functional Ran cycle (with no RanGTP present, as mentioned earlier), which is required for the release of Impβ from the NPC by continuously providing RanGTP in the nucleus (Kutay et al, 1997).

The observed differences for Impβ and NTF2 can be reasoned by the size of these proteins and the number of their FG-Nup–binding sites. NTF2 is a small homodimer (~29 kD) with two FG-Nup–binding sites (Bullock et al, 1996; Bayliss et al, 2002) and is probably optimized to slip through the transport barrier quickly to maintain the RanGDP: RanGTP concentration gradient. Impβ, on the other hand, is significantly larger (~100 kD) and has many more FG-Nup–binding sites (~9) (Isgro & Schulten, 2005). Therefore, due to its larger volume, it should encounter a greater free energy barrier to penetrating the transport barrier (relative to NTF2), as there will be an entropic penalty to limiting the FG-Nups' configurational space, and a further penalty for disrupting FG-Nup·FG-Nup interactions. However, these effects are compensated by its multiple binding sites, such that even with weak individual FG-Nup·Impβ binding interaction strengths (Milles et al, 2015), the overall binding avidity is relatively strong. In our experiments, this translates to a change in the various transport barrier structures (Fig 3), and to their nanomechanical properties (Fig 4). In a physiological context, this suggests that Impβ (with its many binding sites) cross-links the FG-Nups to create a meshwork that occludes the entire channel, thereby making the transport barrier more selective (Fig 5)—that is, it may be considered an "intrinsic" component of the barrier (Jovanovic-Talisman et al, 2009; Kapinos et al, 2014, 2017; Lowe et al, 2015; Wagner et al, 2015). This strongly suggests that transport events require that NTRs are—at least in part—exchanged between the transport barrier and the solution, i.e., that it is rather costly to release a cargo·NTR complex from the FG-Nups without substitution by other NTRs (or by other cargo·NTR complexes).

In summary, these high-resolution AFM results complement recent structural studies (Bui et al, 2013; Eibauer et al, 2015; Von Appen et al, 2015; Kim et al, 2018) by highlighting structural variability that is otherwise lost by ensemble averaging. The observed variability in the NPC lumen is consistent with a functional scenario in which FG-Nups can transition between different (meta-)stable collective conformations to facilitate transport, enabling both the passage of large cargoes and the fast resealing of the barrier. Our results also highlight differences between various NTRs and their interactions with the NPC. They emphasize the relatively strong binding of Impβ and its possible effects on maintaining the transport barrier and on the mechanism of receptor (and thus also cargo) binding and release during nucleocytoplasmic transport.

# Materials and Methods

### NE preparation for AFM imaging

Oocytes were stored in modified Barth's solution (88 mM NaCl, 15 mM Tris, 2.4 mM NaHCO$_3$, 0.82 mM MgCl$_2$, 1 mM KCl, 0.77 mM CaCl$_2$, and U/100 μg penicillin/streptomycin, pH 7.4) at 4°C for a maximum of 3 d. Before isolation, oocytes were transferred to a petri dish—previously treated with BSA—containing nuclear isolation buffer (NIM), composed of 17 mM NaCl, 90 mM KCl, 10 mM MgCl$_2$, 10 mM Tris, and 1.5% wt. polyvinylpyrrolidone (PVP—to mimic the densely packed macromolecular environment of the cytosol and prevent the nuclei from swelling after isolation), pH 7.4. Using tweezers, an oocyte was pinned down and gently pierced just above the equator in the animal pole (if done correctly, the nucleus should begin to burst out of the oocyte). The nucleus was then gently pushed out of the oocyte by pipette aspiration, cleaned by pipette aspiration, and transferred to a new petri dish—also pretreated with BSA—containing NIM (1.5% PVP wt., pH 7.4), where it can be stored on ice for up to ~30 min. After isolating a nucleus, it was then placed in NIM buffer without PVP for 2 min, causing the nucleus to swell and the NE to detach from the chromatin. Once swollen, the nucleus was adsorbed onto a glass coverslip treated with poly-l-lysine in NIM buffer (without PVP, pH 7.4). Sharpened glass capillaries (made by glass

pulling pipettes over a flame), which had been treated with BSA, were used to tear the NE and gently spread it over the glass substrate (trying to expose both areas of cytoplasmic and nucleoplasmic membrane to the buffer). The NIM buffer was exchanged several times (>6) with NIM (PVP 8% wt., pH 7.4), and the sample was incubated at 4°C overnight in a humid environment to prevent loss of buffer—this was to help the membrane stably adsorb onto the glass substrate. The following morning, the sample was washed several times (>6) with the imaging buffer (either import buffer: 20 mM Hepes, 110 mM $CH_3COOK$, 5 mM $Mg(H_3COO)_2$, 0.5 mM EGTA, pH 7.4; or NIM without PVP, pH 7.4). No chemical fixation was used at any point and the sample was kept on ice at all stages of preparation, before transfer to the AFM instrument.

### Atomic force microscopy

A Dimension Icon (Bruker) was used for all force-spectroscopy experiments, and to produce the images displayed in Figs 1A–H and S3; all other images were acquired using a Dimension FastScan (Bruker). MSNL-E (Bruker) probes with a silicon nitride cantilever (nominal $f_0$ in air = 38 kHz; nominal k = 0.1 N/m) and a silicon tip (nominal radius = 2 nm) were used for all force-spectroscopy experiments, both in PeakForce Quantitative Nanomechanical Property Mapping mode (PeakForce QNM) and Force Volume mode. All force-spectroscopy experiments were conducted in NIM buffer (pH 7.4) at room temperature. An MSNL-F (Bruker; nominal $f_0$ in air = 125 kHz, nominal k = 0.6 N/m, nominal tip radius = 2 nm, and tip half opening angle of ≤20°—the full tip shape is defined with ~20° half opening angle, with additional sharpening at the end of the tip; see manufacturer's specifications) was used to produce the images shown in Figs 1A–G and S3, in PeakForce QNM mode at 2 kHz, in NIM (pH 7.4), at room temperature. A ScanAsyst-Fluid-HR (Low k; Bruker; nominal $f_0$ in fluid = 25 kHz, nominal k = 0.05 N/m, nominal tip radius = 1 nm, and tip half opening angle of ≤20°) was used to produce the image shown in Fig 1H (again in Fig S4), using PeakForce QNM mode at 2 kHz, in NIM (pH 7.4), at room temperature. And finally, a FastScan-D (Bruker; nominal $f_0$ in water = 110 kHz, nominal k = 0.25 N/m, nominal tip radius = 5 nm, and tip half opening angle of ≤20°) was used to produce the images shown in Figs 1I–L, 3, and S7, in import buffer (pH 7.4), at room temperature. The images shown in Figs 1I–L and 3 were captured using PeakForce Mapping mode at 8 kHz, whereas the images shown in Fig S7 were captured using Tapping mode.

For all high-resolution imaging experiments (Figs 1, 3, S1, S2, S3, and S4), the samples were imaged at minimum force, that is, such that the contact region of the force curves only just stably rose above the force baseline.

For all imaging and force experiments involving exogenous reagents, the tip was retraced ~500 nm (using the Piezo scanner), the reagent was injected, and the tip was brought back into contact with the surface. Imaging was then conducted in the presence of the reagents (Figs 3A–D, J, K, and 4A–D). The only exceptions are the addition of $hs$Imp$\beta$ to the nucleoplasmic side of the membrane (Fig 3I) and the addition of $hs$Imp$\beta$ to the cytoplasmic side of the membrane followed by washing steps (Fig S7). For these experiments, the tip was withdrawn, the protein was injected, the system was left to incubate (15 min for the data shown in Fig 3I and 25 min

for the data shown in Fig S7), and then the buffer was washed (>5 times). The tip was then re-engaged onto the sample surface with negligible concentrations of $hs$Imp$\beta$ in the imaging buffer (import buffer).

For all force-spectroscopy experiments, a thermal tune was conducted on each cantilever before starting an experiment, to obtain an estimate of its spring constant (the deflection sensitivity of a previously calibrated cantilever from the same batch was used for this). A PeakForce Setpoint equal to ~300–400 pN could then be assigned for all experiments. In PeakForce QNM mode, the Sync Distance QNM parameter was optimized on glass before beginning the experiment (this parameter cannot be optimized on soft materials with a time-dependent response to deformation, such as the NE), and the lift height parameter was frequently adjusted during imaging (it was estimated as being the interaction distance of the force curve) to update the background subtraction algorithms and account for cantilever interactions with the substrate. Images were captured at a resolution of ~3 nm pixel$^{-1}$.

At the end of all force-spectroscopy experiments (in both PeakForce QNM and Force Volume), the deflection sensitivity of the cantilever was calculated by ramping it into glass at a high force such that the deflection of the cantilever is linear with respect to Piezo extension. The gradient of this response was then recorded. This process was repeated 3 times, and the average taken—this value was used as the deflection sensitivity. The spring constant of the cantilever was then recalculated using the thermal tune method. It is important to calibrate the cantilever at the end of the experiment rather than at the beginning as the tip is sometimes damaged when ramping into the surface.

### NTRs, the Ran mix, and the energy mix

The Ran mix (1×) consisted of RanGDP 2 $\mu$M, RNA1p 0.2 $\mu$M, and RanBP1 0.2 $\mu$M (Fassati et al, 2003). NTF2 was prepared as previously described (Kutay et al, 1997). His-tagged Ran, Rna1p, and RanBP1 were prepared as previously described (Görlich et al, 1994; Kutay et al, 1997) with minor modifications. Briefly, PC2 E. coli strains (BL21 (DE3), DendA::Tc$^R$, T1$^R$, pLysS) were grown at 30°C in Luria Bertani medium supplemented with 120 $\mu$g/ml ampicillin to an $A_{600nm}$ of 0.9–1.0, after which the temperature was reduced to 28°C and protein expression was induced with 2 mM IPTG for 3 h. Bacterial pellets were resuspended in ice-cold core buffer (50 mM Hepes, pH 7.0, 5 mM Mg acetate, 100 mM NaCl, and 5 mM $\beta$-mercaptoethanol) supplemented with 1 mM PMSF and disrupted by sonication. Lysates were cleared by centrifugation at 27,000 g for 30 min at 4°C and supernatants incubated with NiNTA resin in the presence of 20 mM imidazole; after washing in core buffer containing 20 mM imidazole, His$_6$-tagged proteins were eluted in core buffer supplemented with 200 mM imidazole. The proteins were further purified by gel filtration over a HiLoad 16/60 Superdex 200 column in 150 mM NaCl, 25 mM Tris–HCl, pH 7.4, and supplemented with 5 mM DTT. Ran was charged by incubation on ice for 30 min in the presence of 10 mM EDTA and 2 mM GDP, after which 25 mM $MgCl_2$ was added. The samples were buffer-exchanged (Hi-Trap desalting column) against import buffer (20 mM Hepes–KOH [pH 7.3], 110 mM potassium acetate, 5 mM magnesium acetate, and 0.5 mM EGTA). Proteins were supplemented with 8% sucrose, flash-frozen in liquid

nitrogen and stored at −80°C. The energy mix (1×) consisted of ATP 1 mM, GTP 1 mM, creatine phosphate 2 mM, and creatine phosphokinase 40 U/ml, pH 7.4. The recombinant human importin-$\alpha$ subunit was purchased from Novoprotein (catalogue number: CE62) and the recombinant human importin-$\beta$ subunit was purchased from Flarebio, CusAb (catalogue number: CSB-EP622929HU). An SDS–PAGE with Coomassie staining was run to check for the purity of the proteins (Fig S8).

# Further methods

### Data analysis protocols

All data analyses were done using MATLAB (MathWorks).

### *Size distribution of NPCs*
The image file is loaded into MATLAB, and the width of the image (fast–scan axis) is entered manually in nm. A first-order plane background subtraction is applied to the image. Circles are found within the image using the MATLAB routine imfindcircles to identify all NPCs (the sensitivity parameter was optimized to best find all pores). All identified NPCs are cropped, and another first-order plane background subtraction is applied to each NPC individually, this time using the top 50% of height data to identify the rim of the pore. Each NPC is then rotationally symmetrized, and the resultant height profile is filtered (Savitzky–Golay method [Savitzky & Golay, 1964]). The peaks in the filtered height profile are then found—this should correspond to the highest point of the NPC scaffold structure. If several peaks are identified, the first peak found at a radial position less than 60 nm from the centre of the NPC is used. The distance from the centre of the NPC to the peak is defined as the NPC radius. The radial values are collated from several images and displayed as a histogram.

### *Nanomechanical characterization of PeakForce QNM data*
The PeakForce QNM data are loaded into MATLAB, and the spring constant and deflection sensitivity for the relevant cantilever are entered manually. A first-order plane background subtraction is applied to each image. The force curves are converted from deflection (V) versus Piezo-z position (nm) to deflection (nm) versus Piezo-z position (nm), using the deflection sensitivity (nm V$^{-1}$). The tip–sample separation (nm) is then calculated using the deflection of the cantilever (nm) as a function of Piezo-z position (nm). Finally, they are converted to force (nN) versus tip–sample separation (nm) using the spring constant (N m$^{-1}$), calculated using the thermal tune method (see the Atomic force microscopy section). The contact point of each force curve is determined (see Fig S9), and the Hertz model is applied from the contact point up to either an indentation of 20 nm, or until the end of the force curve, to give the effective Young's modulus ($E_{eff}$). If the contact point could not be determined, the force curve is removed from the analysis. The centre of each NPC in the image is defined manually. All defined NPCs are then cropped, along with their force curves. For each NPC, the height data, $E_{eff}$ values, and force curves are aggregated based on their radial position from the centre of the pore (the innermost radial bin has a radius of 5 nm; all following concentric circles are in

4-nm intervals). After completing this for all images within one experiment, all files are collated into one data structure. The indentation values for all force curves are determined—if they are very small (<5 nm) or very large (>75 nm), this is considered as incorrect contact point determination, and the force curves are removed. Furthermore, if the $E_{eff}$ values are outside the range $0.1 \leq E_{eff} \leq 10$ MPa, these are considered anomalous (indicating a lack of contact between tip and sample or a sudden jump in the force curve, respectively), and are therefore removed along with the relevant force curves. All remaining force curves, height data, and $E_{eff}$ values are then collated based on their radial position. The height data and $E_{eff}$ values are averaged, and their SD is calculated—this gives the rotationally averaged height and $E_{eff}$ profiles (see Fig S5). Force curves are then averaged based on their radial position. This is done by binning force data into 1-nm-sized bins, and next, averaging the bins. Furthermore, only bins for which 80% of force curves contributed are kept. This removes behaviour at each extreme end of the force curves, which would be prone to artefacts because of averaging from a relatively small subset of data. The negative of the first derivative of the averaged force curves is calculated to produce the stiffness curves. These are then positioned onto an intensity map based on both their radial position from the centre of the NPC, and their height—calculated using the rotationally averaged height profile. This produces the stiffness heatmaps (see Figs 4, S5, and S10). This rotational averaging procedure is carried out on the ensemble averaged data for all NPCs and on each NPC individually.

### *Cross-correlation averaging*
The image file is loaded into MATLAB, and the width of the image (fast–scan axis) is entered manually in nm. A first-order plane background subtraction is applied to the image. Circles are found within the image using the MABLAB routine imfindcircles to identify all NPCs (the sensitivity parameter was optimized to best find all pores). All identified NPCs are cropped, and another first-order plane background subtraction is applied to each NPC individually, this time using the top 50% of height data to identify the rim of the pore. This both aligns each pore horizontally and sets them all to the same height. The first pore recognized by imfindcircles is used as the first template for the cross-correlation averaging. The template is masked such that only image data within an 80-nm radius of the centre of the cropped image is kept for the correlation—this should contain the NPC and ignore background NE (and neighbouring NPCs). A second pore is then compared against the template (this is also masked such that only data within an 80-nm radius of the centre is kept) using the sum of absolute differences method (SAD), that is, the absolute difference in height between each overlapping pixel is taken, and the sum of all differences saved. The image is then rotated 5° and the SAD score is calculated and saved. This is repeated until the image has been rotated 45° (the protocol limits rotation to 45° as eightfold rotational symmetry of the NPC is assumed). The rotation of the image corresponding to the lowest SAD score is defined as having the greatest correlation with the template and is therefore averaged with the template to create a new template. This process is repeated until all pores have been averaged. The averaged image is then rotationally symmetrized to produce the plots shown in Fig 3E–H.

### NPC size distributions: sevenfold and ninefold symmetrical NPCs

As shown in the histogram in Fig S1, of the 583 NPCs, 22 have a recorded diameter of less than 39 nm and 14 have a diameter greater than 49 nm. However, after one-by-one visual inspection of each of these NPCs, it could be seen that some had inaccurately designated radial values because of noise in the image—these were therefore discounted and the final counts are reported as 16 having a radius less than 39 nm and 10 having a radius greater than 49 nm.

Furthermore, all confidence intervals in the text corresponding to scaffold diameter size are reported as ±4 nm. This is from the rotational symmetrizing procedure applied to each NPC. After the centre of each NPC has been defined, concentric circles are drawn up, starting with a radius of 5 nm, then 9 nm, and then 13 nm (etc.), increasing in size by 4 nm. All height data within each range are averaged to produce the rotationally symmetrized plots. Therefore, the true peak-to-peak distance (corresponding to the pore diameter), is anywhere within this range—hence, the confidence interval of ±4 nm (SEM was calculated as ±0.4 nm for cytoplasmic NPCs, $n$ = 583, and as ± 0.8 nm for nucleoplasmic NPCs ± 0.4 nm, $n$ = 282, but these values are artificially small).

### Validating fast force-spectroscopy methods: PeakForce QNM

Traditionally, when acquiring force data simultaneously with the height data of a sample, Force Volume mode was used. This method ramps the AFM cantilever linearly with time and has good force sensitivity; however, data acquisition is slow (usually ~10 force versus distance curves are collected per second in solution [Bestembayeva et al, 2015]). Recently, force-spectroscopy methods that acquire data several orders of magnitude faster than Force Volume have been developed—one such method is PeakForce QNM (Bruker). This method drives the cantilever in a sinusoidal manner. It is faster, but because of a larger measurement bandwidth, it also produces noisier force curves. Therefore, to validate that this new technique, PeakForce QNM, was sensitive enough to discern between the different soft materials comprising the NE (and NPC), it was compared directly with results from Force Volume mode.

Fig S5 displays the results from ensemble-averaged nanomechanical data sets of rotationally averaged NPCs from both the cytoplasmic and nucleoplasmic faces, in both Force Volume and PeakForce QNM modes (see the Nanomechanical characterization of PeakForce QNM data section and Fig S10 for information on the analysis protocol). Force Volume data on the cytoplasmic side of NPCs render a stiffness heatmap (Fig S5A, top) showing increased stiffness at the cytoplasmic ring structure and in the central transport channel. These data are in perfect agreement with previously published results which were obtained using different analysis scripts written in Mathematica (Bestembayeva et al, 2015). The rotationally averaged effective Young's moduli ($E_{eff}$; Fig S5A, bottom)—which is calculated from each force curve individually—renders qualitatively the same pattern, that is, increased elastic response in the centre of the transport channel and at the scaffold ring structure. Fig S5B shows the results from the same experiment; but this time, the data were acquired using PeakForce QNM (2 kHz) mode. Both the stiffness heatmap and rotationally averaged $E_{eff}$ give qualitatively the same results as in Fig S5A, that is,

increased stiffness and elastic response in the centre of the transport channel and at the scaffold ring structure. However, the $E_{eff}$ values rendered from PeakForce QNM mode are larger (by a factor of two to three) than the values given by Force Volume. This is perhaps partly due to the higher velocity of the AFM tip in PeakForce QNM mode (as compared with Force Volume mode), generating a different viscoelastic response from the soft material (the NPC), but also due to the background subtraction algorithms used to obtain a force curve from the sinusoidal drive of the cantilever. If the background subtractions are incorrect, or if the imaging conditions evolve during the experiment (thereby making the background subtractions incorrect), the absolute values given by the Hertz model are incorrect. Furthermore, the Hertz model is dependent upon the contact point determination (see Fig S9), which is also affected by noise within the force curves. However, the results from within one experiment can be used to look at relative differences in the $E_{eff}$. Fig S5C and D shows the results from the nucleoplasmic face of the NPCs: Force Volume (Fig S5C) renders the greatest stiffness and elastic responses from the scaffold structure, but reduced responses from the transport channel. This is because the AFM tip is interacting with the moveable nuclear basket. Again, the results are qualitatively reproduced by PeakForce QNM mode (Fig S5D).

We conclude that PeakForce QNM mode qualitatively reproduces the results obtained from Force Volume, and therefore has the force-sensitivity to elucidate changes in the nanomechanics of soft materials at the nanometre length- and picoNewton force scales. However, because of the evolving background subtractions, and greater noise within each force curve (which makes accurate contact point determination more difficult), it is not considered a completely quantitative technique: relative changes within one experiment can be compared, but differences in absolute values between experiments cannot.

### AFM image processing

All images were flattened in NanoScope Analysis 1.7 (Bruker) (usually a first or a second-order plane background subtraction, dependent upon the image). For presentation of the data that resulted from our analyses, a 3-pixel (~10 nm) Gaussian filter was then applied to the images (the line profiles shown in Fig S2 are from the flattened image after filtering, and all results shown from the MATLAB analysis protocols did not use filtered images). Once flattened and filtered, a false colour scale was applied using Gwyddion.

## Supplementary Information

## Acknowledgements

The authors acknowledge Joseph Beton for his assistance with running the SDS–PAGE (Fig S8) and Aizhan Bestembayeva and Sofya Mikhaleva for

assistance with preliminary experiments. This work was supported by the UK Biotechnology and Biological Sciences Research Council (BB/J014567/1).

## Author Contributions

GJ Stanley: conceptualization, data curation, formal analysis, validation, investigation, visualization, methodology, and writing—original draft, review, and editing.

A Fassati: conceptualization, resources, supervision, funding acquisition, methodology, and writing—original draft, review, and editing.

BW Hoogenboom: conceptualization, software, supervision, funding acquisition, methodology, project administration, and writing—original draft, review, and editing.

## Conflict of Interest Statement

The authors declare that they have no conflict of interest.

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
