## [Reviewer comments · Life Science Alliance]

Atomic force microscopy reveals structural variability amongst nuclear pore complexes

George Stanley, Ariberto Fassati, and Bart Hoogenboom

DOI: 10.26508/lsa.201800142

Review timeline:

Submission Date:	27 July 2018
Editorial Decision:	27 July 2018
Revision Received:	8 August 2018
Editorial Decision:	9 August 2018
Revision Received:	10 August 2018
Accepted:	13 August 2018

Report:

(Note: Letters and reports are not edited. The original formatting of letters and referee reports may not be reflected in this compilation.)

Please note that the manuscript was previously reviewed at another journal and the reports were taken into account in inviting a revision for publication at *Life Science Alliance* prior to submission to *Life Science Alliance*.

1st Editorial Decision

27 July 2018

Thank you for transferring your manuscript entitled "Atomic force microscopy reveals structural variability amongst nuclear pore complexes" to Life Science Alliance. The manuscript was assessed by expert reviewers at another journal before, and the reviewer reports have been transferred to us by the journal editors.

The reviewers noted that your work corroborates and extends previous findings on the NPC, but that the physiological relevance of the reported observations remains rather unclear at this stage. They also noted that functional analyses are currently lacking. All reviewers pointed out that the technical quality of your work is excellent. Based on these assessments obtained elsewhere, we would like to invite you to submit a revised version of your manuscript for publication in Life Science Alliance. The physiological relevance does not need to get addressed. Please provide, however, a full point-by-point response and accordingly text changes/figure improvements to address the technical concerns raised by the reviewers and to clarify aspects of your work that could get currently mis-interpreted. Furthermore, the data should get re-discussed and interpretations toned down to address the specific concern voiced by reviewer #2 and #3. Should you have additional data at hand (eg. to address reviewer #2's comment on mutant importin beta), we think that incorporation of such data would strengthen your manuscript tremendously. Please let us know in case you would like to discuss individual revision points further.

REFeree REPORTS OBTAINED DURING PEER REVIEW ELSEWHERE

Referee #2 Review

Report for Author:

The structural, functional, and biophysical properties of the nuclear pore permeability barrier remain a significant unsolved problem. Developing strategies to investigate this barrier is complicated by the dynamic and heterogeneous nature of the intrinsically disordered FG-nups in combination with the nuclear transport receptors that both carry cargos through the pore and seemingly are fundamental components of the barrier itself. The authors probed the structural and physical properties of nuclear pore complexes (NPCs) in unfixed *Xenopus* nuclear envelopes using high-resolution fast AFM. They have two main findings. First, the NPCs have variable diameters,

which is interpreted to imply the presence of pores with 7- and 9-fold rotational symmetry (~1-3% of total for each) in addition to the anticipated, and well-known, 8-fold symmetry. And second, nanomechanical measurements of stiffness and dynamic changes reveals that the transport receptor importin beta substantially swells the barrier, providing in situ evidence that the structural, and likely the functional, properties of the permeability barrier are modulated by transport receptors. Their quantitative findings will undoubtedly be important for developing more detailed models, particular those based on simulations. The writing is overall very good and the manuscript is easy to read.

NPCs exceeding 8-fold rotational symmetry have been observed earlier in fluorescence (correctly referenced) and EM experiments (add Hinshaw & Milligan, *J Struct Biol*, 2003 141:259). While this certainly suggests the potential for somewhat different transport activities, I am not aware of any evidence that there is an active mechanism to control stoichiometry, and thus, altered stoichiometries could simply be assembly errors. It is not clear that altered stoichiometries must have physiologically important and significant functional consequences despite different structures, so I would tone down the rhetoric a bit in numerous places. Nonetheless, such questions are important for the future, and their approach provides an avenue to pursue this. Are they able to generate averaged images to more clearly validate the different rotational symmetries?

The addition of importin beta clearly changes the structure and mechanical properties of the permeability barrier (which can be defined here as the stuff within the central pore). It would be interesting to know how many importin beta molecules are present in each NPC, but this is undoubtedly beyond the scope of the present work. It is not clear to me what the basis is for the statement that the filling of the NPCs with importin beta is 'irreversible' (p. 9) - did they try to wash it out? If not, why not? I don't understand the statement that their "measurements were taken in the absence of a functional Ran cycle" (p. 9) - didn't they add RanGTP and NTF2? Why doesn't the RanGTP release the importin beta from the FG-nups? Alternately, why is the as-isolated pore not filled with importin beta? If importin beta is irreversibly bound, what does that say about the physiological relevance of what they observe? Some comment/clarification/more balanced presentation would be useful here.

Minor Points:

- 1) p.2 - Some RNPs (and viruses) can escape the nucleus by nuclear blebbing. It is therefore misleading to indicate that all nuclear entry and exit occurs by NPCs.
- 2) p.4 - The finding that some NPCs seem to be connected on the cytoplasmic side is consistent with the finding that some mRNAs exhibit cytoplasmic scanning after export (Smith, *J. Cell Biol*, 2015 211:1121). This is worth noting.
- 3) The English could be cleaned up in a few places, but presumably these will be caught during revision. On p.5 - "To better interpret the AFM images, it would..." could be replaced by "Better interpretation of the AFM images would require..."
- 4) Some description of the AFM tip characteristics (diameter, length, shape, interactions with proteins) in the main text would be useful so that the reader has some idea of how these might influence resolution. The difference between trace and retrace images (Expanded View Figure 2) raises the question about what affect this has on resolution and whether the images shown are an averaged combination of such images. Some clarity here would be helpful.
- 5) p. 7 - It is not clear why the Knockenhauer and Schwartz review is used for the permeability exclusion limit. Better would be Timney *J. Cell Biol* 2016 215:57. The second time this is discussed (p. 9) is a bit repetitive.
- 6) p.9 - I don't understand the argument that NTF2 is significantly smaller than importin beta. NTF2 forms a dimer, and if it binds two molecules of RanGTP (which is present), the total molecular weight would be ~80 kDa.
- 7) p. 9 - I don't understand the concept, nor the basis, for the last sentence of the middle paragraph ("In a more functional context,...").

- 8) Fig. 1 - Is there any reason to rule out that some of the material seen in NPCs is cytoplasmic filaments that have bent back into the pore? After all, they argue that external connections between NPCs are cytoplasmic filaments...
- 9) Fig. 2 - The figure panels are low resolution, particularly the lower panels. The identifying shapes cannot be distinguished. The stiffness maps are noisy and it is not entirely clear what the message is (particularly on the left). Tightly cropped images might aid in comparison. Here and in all AFM figures, it would be useful to indicate the scale in nanometers (or MPa) on each heat bar. I realize that the scales are different for the two sets of images in this Figure, but this is precisely why multiple scale bars should be used. Here and in Expanded View Figure 5, the blank areas (membrane) exhibits some of the stiffest regions. I wonder if some comment can be made on this.
- 10) Expanded View Figure 3 - There is no scale bar.
- 11) Supplementary Figure 1 - What are the criteria for identifying actin and intermediate filaments based on diameters? The more traditional numbers are 7 and 10 nm, respectively, so the justification for actin identification is not clear. There seem to be better places for the line scan in this figure.
- 12) Supplementary Figure 3 - It is unclear what is meant by, and the rationale for, "The standard deviation of each segment is determined and the smallest is taken".
- 13) Supplementary Figure 4 - It is unclear why there is no peak in the center of the pore in H like there is in C and E.

Referee #2 Review

Report for Author:

In their manuscript "Atomic force microscopy reveals structural variability amongst nuclear pore complexes" by Stanley et al., the authors describe an improved high-resolution atomic force microscopy method for the analysis of NPCs (nuclear pore complexes) from *Xenopus laevis* oocytes. They describe heterogeneity of NPCs at different levels: first, the dimensions can vary and this variation can be assigned to 7-, 8- or 9-fold rotational symmetry of the NPC. This observation by AFM is interesting. It has been made, however, before (e.g. Hinshaw and Milligan, 2003; Loschberg et al., 2014) using different methods and its significance is not further investigated. Second, the authors describe variable substructures - e.g. central protrusions (perhaps cargos in transit), local densities within the lumen of the NPC, which may result from connecting FG-Nups, or cytoplasmic filaments connecting two NPCs. Again, the significance of these structures is not further analyzed. In addition, filaments are observed on the nuclear side of the NPC, most likely components of the nuclear lamina. Differences in densities of the filamentous meshwork are suggested to result from "different mechanical strain applied to the nuclear envelope during sample preparation". This interpretation is not further tested. Next, the authors perform nanomechanical analyses of individual pores. Differences in the mechanical properties are observed, correlating with the morphological features of the NPC. The experimental setup allows a comparison of mechanical and morphological features of individual NPC before and after the addition of certain components. The authors then analyze the effects of RanGDP, NTF2 and importin beta. They nicely show that importin beta leads to morphological changes (resulting from binding of the protein to nucleoporins) and an increased stiffness. Together, the technical quality of the paper is very high, but the interpretation of the results remains rather speculative. In the abstract, the authors conclude "that FG-nup cohesiveness is tuned to allow collective rearrangements at little energetic cost". What is the point of tuning at the level of individual pores (especially for an oocyte at a defined developmental state) and how could such tuning be accomplished? No evidence for differences in cohesiveness is presented. At this point in time, where there are still alternative transport models in the literature (e.g. cohesive vs. non-cohesive Nups), the AFM methodology could be used to perform a more functional analysis (e.g. can the importin beta effect be reversed by the addition of RanGTP? Would importin beta with mutations in some of the FG-binding sites behave differently? Would hydrophobic reagents like cyclohexanediol (see Ribbeck and Gorlich, 2002) affect the stiffness of the

NPC/ the material in the lumen of the NPC). Without such an analysis, we are left with a rather descriptive manuscript that might be more appropriate for a more specialized journal.

Minor points:

page 12: explain/cite the Savitzky-Golay method

page 37: what is the band at 50 kDa in the Ran-mix?

Referee #3 Review

Report for Author:

In this study, the authors use high-resolution atomic force microscopy (AFM) to study individual nuclear pore complexes (NPCs). The authors are particularly focussing on the central pore region of NPCs and reveal structural diversity, which they attribute to rearrangements of the FG-Nup network. While the provided AFM images are of remarkable high resolution, the authors' conclusions appear largely over interpreted.

1. In Figure 1, the authors show cytoplasmic and nucleoplasmic views of the nuclear envelope. In the overview image showing the nucleoplasmic side (Fig. 1H), it can be seen that the nuclear basket, as it is known, is highly flexible and variable and that some NPCs appear to lack basket. Taken this into account and what is known from electron microscopy studies (Aebi lab, Allen lab, for example), the interpretation of the images shown in Fig. 1b-g, are in my opinion over or mis interpreted. Rather than reflecting changes in the pore lumen itself, the images show cytoplasmic views of the nuclear basket. Whatever the correct interpretation may be, as the authors themselves have written, a proper interpretation of the AFM images requires molecular identification. Without a molecular identification (by immune-labelling, for example), all conclusions are purely speculative.

2. The authors observed NPCs with rotational symmetry that deviates from the well-known 8-fold rotational symmetry. This is not new and at the very most a confirmation of previous EM studies by Hinshaw et al. in 2003 (JSB 141:259-268).

3. The authors claim that they have carried out "high-throughput" nanomechanical characterisation of the nuclear envelope. What is meant by high-throughput here? Numbers of analysed nuclear envelope and nuclear pores are not given, except for Extended Figure 1. Are the authors equalising high-speed with high-throughput?

4. According to the journal guideline, Results and Discussion should be combined.

1st Revision – authors' response

8 August 2018

Referee #1:

The structural, functional, and biophysical properties of the nuclear pore permeability barrier remain a significant unsolved problem. Developing strategies to investigate this barrier is complicated by the dynamic and heterogeneous nature of the intrinsically disordered FGnups in combination with the nuclear transport receptors that both carry cargos through the pore and seemingly are fundamental components of the barrier itself. The authors probed the structural and physical properties of nuclear pore complexes (NPCs) in unfixed Xenopus nuclear envelopes using high-resolution fast AFM. They have two main findings. First, the NPCs have variable diameters, which is interpreted to imply the presence of pores with 7- and 9-fold rotational symmetry (~1-3% of total for each) in addition to the anticipated, and well-known, 8-fold symmetry. And second, nanomechanical measurements of stiffness and dynamic changes reveals that the transport receptor importin beta substantially swells the barrier, providing in situ evidence that the structural, and likely the functional, properties of the permeability barrier are modulated by transport receptors. Their quantitative findings will undoubtedly be important for developing more detailed models, particular those based on simulations. The writing is overall very good and the manuscript is easy to read.

Response: We thank the referee for this positive assessment of our manuscript. In response to these comments, we would like to point out that the third main finding of our manuscript is the heterogeneous appearance of the NPC central channel. We consider this an important addition to the currently available structural information (see references in Introduction), which is based on ensemble averaging and in which such heterogeneity is masked.

NPCs exceeding 8-fold rotational symmetry have been observed earlier in fluorescence (correctly referenced) and EM experiments (add Hinshaw & Milligan, J Struct Biol, 2003 141:259). While this certainly suggests the potential for somewhat different transport activities, I am not aware of any evidence that there is an active mechanism to control stoichiometry, and thus, altered stoichiometries could simply be assembly errors. It is not clear that altered stoichiometries must have physiologically important and significant functional consequences despite different structures, so I would tone down the rhetoric a bit in numerous places. Nonetheless, such questions are important for the future, and their approach provides an avenue to pursue this. Are they able to generate averaged images to more clearly validate the different rotational symmetries?

Response: We have added the reference to Hinshaw & Milligan in the context of the differently sized NPC scaffolds. Regarding the rhetoric, we presume that this refers to a sentence in the abstract and a sentence at the end of the Discussion, raising the possibility of functional relevance of these altered stoichiometries: We have removed these sentences. Re averaged images to validate the rotational symmetries: Under ideal conditions, our AFM images reveal signatures of the 8-fold rotational symmetry, but not consistently enough for us to reveal the alternate stoichiometries (which are rather rare, as quantified in Expanded View Figure 1) by ensemble averaging.

The addition of importin beta clearly changes the structure and mechanical properties of the permeability barrier (which can be defined here as the stuff within the central pore). It would be interesting to know how many important beta molecules are present in each NPC, but this is undoubtedly beyond the scope of the present work. It is not clear to me what the basis is for the statement that the filling of the NPCs with importin beta is 'irreversible' (p. 9) - did they try to wash it out? If not, why not? I don't understand the statement that their "measurements were taken in the absence of a functional Ran cycle" (p. 9) - didn't they add RanGTP and NTF2? Why doesn't the RanGTP release the importin beta from the FG-nups? Alternately, why is the as-isolated pore not filled with importin beta? If importin beta is irreversibly bound, what does that say about the physiological relevance of what they observe? Some comment/clarification/more balanced presentation would be useful here.

Response: We did attempt to wash importin beta out, but without success. This data has now been included as a new Supplementary Figure 2, and reference has been made to it both in the Results (page 7) and the Discussion (page 9). Furthermore, the text was changed to remove 'irreversible binding of ImpBeta', and replace it with 'strong/stable' binding of ImpBeta. The Methods section has also been adjusted to include the information on the new Supplementary Figure 2 (Methods section 4.2; p. 11).

Regarding the Ran cycle, it is important to clarify that the experiments were carried out on isolated nuclear envelopes. As such, RanGDP cannot be converted into RanGTP because the required nuclear RanGEF, RCC1, is absent. Instead, the energy and Ran mixes added to the experiments promote the generation of RanGDP to better mimic the cytoplasmic compartment. To further clarify this, the text has been amended in the Results section 2.3 (p. 6).

We agree that it would be useful to also characterise importin beta binding to the NPCs after addition of RanGTP, but this turns out to be technically difficult and has – in our hands – thus far not led to conclusive results. Firstly, the addition of RanGTP appeared to cause increasing problems with AFM tip contamination and/or detachment of the nuclear envelope from the substrate, complicating the acquisition of high-resolution AFM data. In addition, it may be that Importin beta is only turned over significantly at the nucleoplasmic periphery (at Nup153 for example, see Lowe et al., eLife, 2015). This would require a more extensive investigation and, also taking into account the technical difficulties mentioned above, is

beyond the scope of this paper.

After isolation, the NPC may be expected to still contain some bound importin beta, but not to the extent as shown here following addition of recombinant importin beta. In particular, after isolation of the nuclei, the RanGTP would still be generated by RanGEF (RCC1) in the nucleus for some time, which may promote the completion of import cycles, without further addition of cargo molecules or NTRs from the cytoplasmic face (as the cytosol has been removed). In addition, the nuclear envelope is detached from the chromatin by allowing it to swell in NIM buffer without PVP. This swelling procedure causes a large influx of buffer through the pores, and may also contribute to a flushing of the pores, with a further depletion of NTRs as a result.

Taking these points into account, we have amended the Discussion to provide a clearer and more balanced presentation, as suggested by the referee.

Minor Points:

1) p.2 - *Some RNPs (and viruses) can escape the nucleus by nuclear blebbing. It is therefore misleading to indicate that all nuclear entry and exit occurs by NPCs.*

Response: This refers to the first sentence of the Introduction, which has been corrected accordingly (removing “all”).

2) p.4 - *The finding that some NPCs seem to be connected on the cytoplasmic side is consistent with the finding that some mRNAs exhibit cytoplasmic scanning after export (Smith, J. Cell Biol, 2015 211:1121). This is worth noting.*

Response: This has now been noted (and referenced) following its observation in the Results section (p. 4).

3) *The English could be cleaned up in a few places, but presumably these will be caught during revision. On p.5 - "To better interpret the AFM images, it would..." could be replaced by "Better interpretation of the AFM images would require..."*

Response: This has been corrected and we have again proofread the manuscript, adjusting the English where appropriate.

4) *Some description of the AFM tip characteristics (diameter, length, shape, interactions with proteins) in the main text would be useful so that the reader has some idea of how these might influence resolution. The difference between trace and retrace images (Expanded View Figure 2) raises the question about what affect this has on resolution and whether the images shown are an averaged combination of such images. Some clarity here would be helpful.*

Response: We have now mentioned tip radius and tip half opening angle at the beginning of the Results section, p. 4, and also in the Methods section (4.2). In discussing the Expanded View Figure 2 on p. 4, we have specified that “*trace and retrace images of the same pores are slightly shifted with respect to each other [the difference mentioned by the referee] due to scanner hysteresis, but otherwise not significantly different*”. Because of this shift, trace and retrace are stored as separate images, and not averaged (but used for consistency checks as presented here). This is the usual AFM image acquisition procedure and has been further clarified in the caption of Expanded View Figure 2.

5) p.7 - *It is not clear why the Knockenhauer and Schwartz review is used for the permeability exclusion limit. Better would be Timney J. Cell Biol 2016 215:57. The second time this is discussed (p. 9) is a bit repetitive.*

Response: The reference has been amended accordingly and the repetition on p. 9 removed.

6) p.9 - *I don't understand the argument that NTF2 is significantly smaller than importin beta.*

NTF2 forms a dimer, and if it binds two molecules of RanGTP (which is present), the total molecular weight would be ~80 kDa.

Response: We respectfully disagree with the referee here. The total molecular weight of a functional NTF2 dimer is ~29 kDa, (see, for example, Bullock et al, 1996) and the SDS-PAGE in Supplementary Figure 3, which shows the monomer as a band between 10 and 15 kDa markers. In the text, we have now emphasised that the ~29 kDa refers to the *total* molecular weight of the NTF2 dimer.

7) p.9 - I don't understand the concept, nor the basis, for the last sentence of the middle paragraph ("In a more functional context,...").

Response: This has been rephrased (see the end of the one-but-last paragraph of the Discussion section).

8) Fig. 1 - Is there any reason to rule out that some of the material seen in NPCs is cytoplasmic filaments that have bent back into the pore? After all, they argue that external connections between NPCs are cytoplasmic filaments...

Response: We have amended the text in the Discussion (p. 8) to make clear that a contribution of cytoplasmic filaments cannot be fully excluded, but is most unlikely given their flexible nature in solution. Regarding the “external connections” discussed in connection with Figure 1, we note that the “spindly protrusions” are only very faint – in the cases where they are visible at all – in the AFM images, again consistent with the flexible nature of the cytoplasmic filaments, and unlike the robust variations in structure as observed amongst different NPC channels.

9) Fig. 2 - The figure panels are low resolution, particularly the lower panels. The identifying shapes cannot be distinguished. The stiffness maps are noisy and it is not entirely clear what the message is (particularly on the left). Tightly cropped images might aid in comparison. Here and in all AFM figures, it would be useful to indicate the scale in nanometers (or MPa) on each heat bar. I realize that the scales are different for the two sets of images in this Figure, but this is precisely why multiple scale bars should be used. Here and in Expanded View Figure 5, the blank areas (membrane) exhibits some of the stiffest regions. I wonder if some comment can be made on this.

Response: The identifying shapes in Figure 2 have been removed and replaced with numbers. The pores have also been tightly cropped and shown as a new panel in the figure. The text has been changed accordingly to make the message of the figure clearer (Results section 2.2; p. 6).

Regarding the various E_{eff} values recorded at the nuclear envelope surrounding NPC, we note that the isolation procedure (see Methods) relies on adsorbing the nuclear envelope onto the underlying substrate. The stiffer appearance of the “membrane” here can be attributed to the proximity of this substrate. This does not (or much less so) apply to the NPC, as its surface is separated from the underlying substrate by tens of nanometers of protein contents. Indeed, as demonstrated in control experiments in previous studies (Bestembayeva et al., 2015), the substrate has no significant effect on the measured nanomechanical properties of the NPC itself.

Regarding scale bars, we have followed the guidelines of the journal, which suggest to define the numerical values in the captions rather than on the scale bars in the figures. We would of course be happy to amend this if deemed appropriate by the editor.

10) Expanded View Figure 3 - There is no scale bar.

Response: The scale bar has been added.

11) Supplementary Figure 1 - What are the criteria for identifying actin and intermediate filaments based on diameters? The more traditional numbers are 7 and 10 nm, respectively,

so the justification for actin identification is not clear. There seem to be better places for the line scan in this figure.

Response: We have used diameters from the EM data in Turgay et al, Nature, 2017. In this paper, actin filaments are reported as having a width of 8 nm, to which one here may add 1-2 nm due to convolution with the AFM tip. In response to the referee's comment, Supplementary Figure 1 has been changed to include more line scans, and the caption changed to explain tip-convolution effects in AFM imaging, which lead to values that are still smaller than expected for intermediate filaments. In addition, the Results section has been amended to read: "Given their widths and lengths, as well as their apparent anchoring to the nuclear envelope, these are likely actin filaments (see Supplementary Figure 1).

12) Supplementary Figure 3 - It is unclear what is meant by, and the rationale for, "The standard deviation of each segment is determined and the smallest is taken".

Response: This is to obtain a measure for the baseline noise, as now specified in the caption of this Supplementary Figure (Supplementary Figure 4 in this revised manuscript). Effectively, the contact point is found by verifying for which indentation the force first significantly exceeds the baseline noise.

13) *Supplementary Figure 4 - It is unclear why there is no peak in the center of the pore in H like there is in C and E.*

Response: This figure (now Supplementary Fig. 5) is intended as a schematic to demonstrate the analysis protocol. As such, the stiffness heatmap shown in H is the averaged 'final result' from 145 characterised NPCs, and not the stiffness heatmap of the one pore shown in C-E. If the stiffness heatmap in H were just of the pore shown in C-E, a peak in the height profile would be expected in the centre. Bestembayeva et al. 2015 contains further technical details on the combined imaging and nanomechanical characterisation, and is referred to in the caption of this figure.

Referee #2:

*In their manuscript "Atomic force microscopy reveals structural variability amongst nuclear pore complexes" by Stanley et al., the authors describe an improved high-resolution atomic force microscopy method for the analysis of NPCs (nuclear pore complexes) from *Xenopus laevis* oocytes. They describe heterogeneity of NPCs at different levels: first, the dimensions can vary and this variation can be assigned to 7-, 8- or 9-fold rotational symmetry of the NPC. This observation by AFM is interesting. It has been made, however, before (e.g. Hinshaw and Milligan, 2003; Loschberger et al., 2014) using different methods and its significance is not further investigated. Second, the authors describe variable substructures - e.g. central protrusions (perhaps cargos in transit), local densities within the lumen of the NPC, which may result from connecting FG-Nups, or cytoplasmic filaments connecting two NPCs. Again, the significance of these structures is not further analyzed. In addition, filaments are observed on the nuclear side of the NPC, most likely components of the nuclear lamina. Differences in densities of the filamentous meshwork are suggested to result from "different mechanical strain applied to the nuclear envelope during sample preparation". This interpretation is not further tested.*

Response: Hinshaw and Milligan 2003 has now been cited in addition to the already included reference to Löscherger et al 2014. In general, our results were analysed, interpreted and tested as far as possible given the nature of the underlying data.

Next, the authors perform nanomechanical analyses of individual pores. Differences in the mechanical properties are observed, correlating with the morphological features of the NPC. The experimental setup allows a comparison of mechanical and morphological features of individual NPC before and after the addition of certain components. The authors then analyze the effects of RanGDP, NTF2 and importin beta. They nicely show that importin beta leads to morphological changes (resulting from binding of the protein to nucleoporins) and an increased stiffness.

Together, the technical quality of the paper is very high, but the interpretation of the results remains rather speculative. In the abstract, the authors conclude "that FG-nup cohesiveness is tuned to allow collective rearrangements at little energetic cost". What is the point of tuning at the level of individual pores (especially for an oocyte at a defined developmental state) and how could such tuning be accomplished? No evidence for differences in cohesiveness is presented.

Response: With hindsight, we believe the word “tuned” is too suggestive and too open for misinterpretation. We have amended this in the manuscript. Specifically, taking note of the caveats as made explicit in the manuscript, the large structural variety in the NPC channel suggests that there are multiple distinct FG-nup arrangements. This is consistent with the prediction that the cohesiveness of the FG-nups is in a range (not too strong, not too weak) that allows transitions between distinct (meta-)stable collective states.

At this point in time, where there are still alternative transport models in the literature (e.g. cohesive vs. non-cohesive Nups), the AFM-methodology could be used to perform a more functional analysis (e.g. can the importin beta effect be reversed by the addition of RanGTP? Would importin beta with mutations in some of the FG-binding sites behave differently? Would hydrophobic reagents like cyclohexanediol (see Ribbeck and Gorlich, 2002) affect the stiffness of the NPC/the material in the lumen of the NPC). Without such an analysis, we are left with a rather descriptive manuscript that might be more appropriate for a more specialized journal.

Response: We agree that there is ample scope for further experiments, in particular the addition of RanGTP. However, this turns out to be technically difficult and has – in our hands – thus far not led to conclusive results. Firstly, the addition of RanGTP appeared to cause increasing problems with AFM tip contamination and/or detachment of the nuclear envelope from the substrate, complicating the acquisition of high-resolution AFM data. In addition, it may be that Importin beta is only turned over significantly at the nucleoplasmic periphery (at Nup153 for example, see Lowe et al., eLife, 2015). This, as well as an analysis in terms of different importin beta mutants (see e.g., older work by Jaeggi et al. 2003, Biophys J: 84, 665–670) would require a more extensive investigation and, also taking into account the technical difficulties mentioned above, is beyond the scope of this paper. The cyclohexanediol suggestion is a priori interesting, but results would be hard to interpret because cyclohexanediol was reported to also cause irreversible release of nups from the NPC (Liashkovic et al., 2012, J. Control. Release 160: 601–608). Regarding functional analysis in general, it is (and remains) one of the main challenges in the field to integrate functional assays with the ability to resolve structures in the NPC channel at ~1 nm resolution.

Minor points:

page 12: explain/cite the Savitzky-Golay method

Response: A relevant citation has now been included.

page 37: what is the band at 50 kDa in the Ran-mix?

Response: It is *S. pombe* Rna1p, the functional homologue of human RanGAP1, which is 386aa and has a mass of 44 kDa (Melchior et al. Mol. Biol. Cell 1993 4:569), giving a total mass of approx. 50kDa with the 6x his tag. The *S. pombe* Rna1p is routinely used to make the Ran mix (see publications by the Görlich lab).

Referee #3:

In this study, the authors use high-resolution atomic force microscopy (AFM) to study individual nuclear pore complexes (NPCs). The authors are particularly focussing on the central pore region of NPCs and reveal structural diversity, which they attribute to rearrangements of the FG-Nup network. While the provided AFM images are of remarkable

high resolution, the authors' conclusions appear largely over interpreted.

Response: As also noted by the referee in comment 1 below (regarding molecular identification), our manuscript explicitly states the caveats that apply to our interpretation. Our interpretation is what we consider the most plausible one given the experimental data and given our knowledge of the system under study and the imaging process. In accordance with good scientific practice, it has been noted well separated from the description of the experimental facts, to avoid confusion and to allow the reader to draw his/her own conclusions about the plausibility of our interpretation.

1. In Figure 1, the authors show cytoplasmic and nucleoplasmic views of the nuclear envelope. In the overview image showing the nucleoplasmic side (Fig. 1H), it is can be seen that the nuclear basket, as it is known, is highly flexible and variable and that some NPCs appear to lack basket. Taken this into account and what is know from electron microscopy studies (Aebi lab, Allen lab, for example), the interpretation of the images shown in Fig. 1b-g, are in my opinion over or mis interpreted. Rather than reflecting changes in the pore lumen itself, the images show cytoplasmic views of the nuclear basket. Whatever the correct interpretation may be, as the authors themselves have written, a proper interpretation of the AFM images requires molecular identification. Without a molecular identification (by immune-labelling, for example), all conclusions are purely speculative.

Response: AFM is a surface technique; at the cytoplasmic surface of the NPCs, it probes relatively stiff structures that are multiple tens of nanometres removed from the more flexible nuclear basket. Unlike EM, AFM simply does not have the ability to “see through” the NPC from the relatively stiff top (cytoplasmic) surface to the bottom (nuclear basket). We refer to our AFM controls that compared the cytoplasmic surface of nuclear envelopes with intact NPCs to nuclear envelopes from which the NPC nuclear basket had been removed: In those experiments, there were no significant differences in either the structure or mechanical properties at the cytoplasmic surface (Bestembayeva et al. 2015). The suggestion of immune labelling is an interesting one, but not very practical since the immune labels would obscure the underlying structure(s) for the AFM.

2. The authors observed NPCs with rotational symmetry that deviates from the well-known 8-fold rotational symmetry. This is not new and at the very most a confirmation of previous EM studies by Hinshaw et al. in 2003 (JSB 141:259-268).

Response: In reporting deviations from the 8-fold symmetry, we had noted and cited the previous report by Löscherberger et al. We welcome the suggestion to also include the prior Hinshaw & Milligan reference and have of course cited it in this revised submission.

3. The authors claim that they have carried out "high-throughput" nanomechanical characterisation of the nuclear envelope. What is meant by high-throughput here? Numbers of analysed nuclear envelope and nuclear pores are not given, accept for Extended Figure 1. Are the authors equalising high-speed with high-throughput?

Response: We indeed use high speed and high throughput interchangeably in the context of these experiments. As implied by the two-orders of magnitude difference in speed (as noted on p. 5), the higher acquisition speed makes it possible to obtain nanomechanical data on a membrane area of 400x400 nm in ~1 minute. To acquire nanomechanical data on the same area at the same pixel density using Force Volume mode takes ~20 minutes. Acquiring a full data set (with tens of NPCs, and taking into account experimental set-up time) therefore takes a day using Force Volume mode (and sometimes several days), whereas several data sets can be recorded in a single day with the newer force-capture mode, PeakForce QNM. To us, this means a noteworthy difference in experimental throughput. The numbers of nuclear pores in our nanomechanical analyses are given in the relevant figure captions (Figure 4, Expanded View Figure 4, Supplementary Figure 5). However, the referee is correct that the number of nuclear envelopes was not given. For each experimental condition stated, these are results from n NPCs of the same nuclear envelope. This has now been made clear in each of the three concerned Figure captions.

4. According to the journal guideline, Results and Discussion should be combined.

Response: We have formatted this revised submission according to the relevant journal guidelines.

2nd Editorial Decision

9 August 2018

Thank you for submitting your revised manuscript entitled "Atomic force microscopy reveals structural variability amongst nuclear pore complexes". We appreciate the introduced changes and would be happy to publish your paper in Life Science Alliance pending final revisions necessary to meet our formatting guidelines.